# Human Papillomavirus Genotypes Infecting the Anal Canal and Cervix in HIV+ Men and Women, Anal Cytology, and Risk Factors for Anal Infection

**DOI:** 10.3390/pathogens12020252

**Published:** 2023-02-04

**Authors:** Laura Conde-Ferráez, Alberto Chan-Mezeta, Jesús Gilberto Gómez-Carballo, Guadalupe Ayora-Talavera, María del Refugio González-Losa

**Affiliations:** Centro de Investigaciones Regionales, Autonomous University of Yucatan, Avenida Itzáes, No. 490 x Calle 59, Col. Centro, Mérida 97000, Mexico

**Keywords:** HPV, HIV, anus, MSM, anal cytology, HPV genotyping

## Abstract

The incidence of anal intraepithelial neoplasias associated with HPV is rising worldwide. In the general population, this pathology is rare, but individuals living with HIV/AIDS are at a significantly higher risk. We aimed to study HPV infection and performed cytological screening to study the epidemiological and behavioral determinants in a group of men and women living with HIV from a region in Mexico with high HIV incidence. This was a cross-sectional study including adults living with HIV/AIDS performed in Merida (Mexico). We invited patients of public HIV/STD clinics and those affiliated with social organizations of people living with HIV to participate in the study. Participants responded to an instrument to assess their risky behaviors and clinical history. Swabs from the anal canal and cervix and anal cytology specimens were obtained by medical staff from women and by self-sampling from men. For the 200 participants, 169 men and 31 women, anal HPV PCR tests resulted in 59.8% positivity (62.6% of men and 45.2% of women), and 17 genotypes were identified. The most frequent high-risk (HR) types for the anal canal were: HPV33 (35.3%), HPV58 (20.6%), HPV66 (18.6%), HPV45 (17.6%), and HPV16 (14.7%). Multiple genotypes were found in over 80% of the participants. Receptive anal intercourse in the previous 12 months, inconsistent condom use, and detectable HIV titers (≥50 cc/mL) were associated with HPV infection (*p* < 0.05). Cytology (smears and liquid-based) identified that 34.6% of the participants had low-grade squamous intraepithelial lesions (LSILs), and 3.5% had high-grade squamous intraepithelial lesions (HSILs). Neither HPV nor lesions were associated with low CD4+ counts (<200 cells/mm^3^, *p* > 0.05). Of the women, 60% were infected in the cervix and 45% in the anal canal, with an agreement of at least one genotype in 90%. The HR-HPV types associated with HSILs were HPV66, 33, 52, 51, 45, 18, and 68.

## 1. Introduction

Human papillomavirus (HPV) is the most common sexually transmitted viral infection worldwide. The WHO has estimated that all sexually active subjects will be infected at some point in their lives. As these are mostly subclinical infections, the relevance of HPV infection is not only due to its high incidence, but also because many genotypes are associated with cancer development. Worldwide, 4.5% of all cancers are associated with HPV infection. The relative contribution of genotypes HPV16/18 and HPV6/11/16/18/31/33/45/52/58 was also estimated to represent 4.5% of all cancers worldwide [1].

Persistent infection with oncogenic HPV genotypes is the main risk factor for developing anal intraepithelial neoplasia (AIN). AIN is an infrequent pathology that has risen in incidence in the last few decades, with people living with HIV/AIDS being at higher risk [2]. Unlike cervical cancer and other HPV-related pathologies, AIN remains understudied, and there are no official international screening programs addressing the populations at risk [3]. The gold standard for diagnosis is high-resolution anoscopy; however, its availability is very limited in most settings [4]. For secondary prevention, it has been proposed that the screening of the at-risk population should include HPV detection and cytology, similarly to cervical disease, in order to detect genotypes with oncogenic potential in a timely manner and identify the presence of AIN precursor lesions [5]. Despite the possibility of performing HPV detection and cytology in many countries, AIN screening programs are still not a reality. The barriers include a lack of trained personnel for sampling, processing, and the interpretation of the results.

HPV infection in HIV patients is reportedly common and presents with multiple genotypes. The genotype distribution varies geographically. In Latin America, studies on HPV infection in the anal canal and associated pathologies are still limited.

We aimed to study HPV infection and perform cytological screening to generate knowledge regarding the anal HPV epidemiology and behavioral determinants in a group of men and women living with HIV from a region in Mexico with high HIV incidence.

## 2. Materials and Methods

This was a cross-sectional study including adults living with HIV/AIDS performed in Merida (Yucatan, Mexico). The project was approved by the Ethics Committee Board of the Research Center “Dr. Hideyo Noguchi” (protocol number record CEI-0014-2016). The internal academic register is CIR-B-2017-0003.

Patients of public HIV/STD clinics were invited to participate through printed fliers, and patients affiliated to four social organizations of people living with HIV/AIDS were invited through in-person group sessions or talks. The inclusion criteria for participation were: adults (>18 years old) with a HIV+ diagnosis who provided informed consent.

Two anal canal samples were obtained from the male participants for a conventional Pap smear and liquid-based cytology. Samples from male participants were obtained with a cytobrush by self-sampling, following instructions provided by the staff. In the case of women, two samples from the anal canal and one from the cervix were obtained with Dacron swabs and a cytobrush, respectively, by trained personnel during pelvic examinations; self-sampling was not adopted by women. Conventional Pap-smear slides from the anal canal samples were obtained and fixed with cytospray on site and, together with liquid-based cytology specimens, were transported to an external laboratory for processing.

After the anal Pap-smear slides were prepared, the remaining anal cells from the sampling device were recovered in saline solution, which was supplemented with an equal volume of absolute ethanol to produce a final 50% ethanol concentration; cervical cells were directly introduced in 50% ethanol at room temperature. All cell samples in ethanol were transported to the virology laboratory at room temperature for processing for the molecular detection of HPV, as described below.

Anal cells from all participants and cervical cells from women were washed with PBS solution to remove the ethanol, and DNA extraction was performed with a DNAeasy Blood and Tissue Kit (Qiagen, Hilden, Germany) following the manufacturer’s instructions.

Human β-globin was amplified as an internal quality control with end-point PCR, using 1 μM oligonucleotides PC04 and GH20 (Saiki et al. (1986), as reported in [6]) in a 25 µL reaction volume with Dream Taq Green Master mix (Thermo Scientific^®^, Waltham, MA, USA); the following cycling conditions were applied: 40 cycles of 95 °C for 30 s, annealing at 55 °C for 30 s and 72 °C for 30 s, and final elongation at 72 °C for 10 min. Amplification products were visualized in agarose gels.

HPV detection was performed by the amplification of the conserved L1 gene region with end-point PCR, using oligonucleotide mix PGMY11/09 [6]. We added 0.2 μM of each primer to Dream Taq Green Master mix (Thermo Scientific^®^), reaching a volume of 50 µL, and subjected the mix to the following cycling conditions: 40 cycles of 95 °C for 1 min; annealing at 55 °C for 1 min and 72 °C for 1 min; and final elongation at 72 °C for 5 min. Amplification products were visualized in acrylamide 8% gels.

Affordable HPV genotyping was performed with multiplex nested end-point PCR following Sotlar et al.’s [7] protocol, which included 18 HPV genotypes: 14 high-risk (HR) and 4 low-risk (LR). Amplification products were visualized in acrylamide 8% gels.

Pap smears and liquid-based cytology specimens from the anal canal were evaluated in an external laboratory by a certified cytopathologist.

Participants responded to a survey instrument to assess their epidemiological risk factors, including sociodemographic information; sexual behavior; self-reported clinical history; tobacco, drug, and alcohol use; CD4+ status; and HIV viral load. Data recovered from the survey and laboratory results were combined and analyzed in SPSS database software using the chi-square test and odds ratio (significance *p* < 0.05). The final database is available from the authors upon reasonable request.

## 3. Results

Because we used a survey instrument that had not been previously validated, an initial pilot was performed including 12 volunteers (eight men and four women) to adjust the items. The same group was allowed to undergo testing in the most comfortable position for anal sampling before the project started (data not shown, will be published separately).

In this project, 204 HIV+ persons volunteered to participate; after 4 were excluded for not adhering to the inclusion criteria, 200 participants were finally included in the study, and all participants provided informed consent.

### 3.1. Description of the Studied Population

The 200 included participants comprised 169 men and 31 women. The age distribution among the participants was between 18 and 72 years, with a mean of 37 years of age.

The occupation data showed that 69.5% (139/200) were employed, 9% (18/200) were unemployed, 9% (18/200) were students, 9.5% (19/200) were homemakers, and 3% (6/200) were retired. Regarding marital status, 80.5% (161/200) were single; 9% (18/200) were married; and 10.5% (21/200) were divorced, widowed, or other.

Regarding tobacco exposure, 33% (66/200) were smokers, and 35% (70/200) indicated passive exposure (cohabitating with a smoker). People who injected drugs represented only 4% of the cohort (8/200).

All participants were sexually active; the average age at first intercourse was 16 years old for men (range 4–30 years), and 15 years old for women (range 6–25 years); young ages may indicate child abuse (data not shown). The age ranges and averages at first intercourse with a male partner were the same for both men and women.

Overall, according to the analysis of sexual practices, 83% (166/200) reported that they had engaged in anoreceptive intercourse, with an average of 41 sexual partners in a lifetime and 5 sexual partners in the last 12 months. Active oral sex was reported by 79.5% (159/200) of all participants, with an average of 11 partners in the last 12 months.

In men, 94% (159/169) reported having sex with men during their lifetime (all sexual practices). Anoreceptive sexual practices were reported by 87% (148/169) of them, with an average of 7 sexual partners in the last 12 months. In the case of women, 42% (18/31) reported anal intercourse, 61% (11/18 of those engaging in anal sex) with 1–2 sexual partners in the last 12 months.

Lifetime consistent condom use during oral sex was reported by only 4.5% (8/179 of the participants engaging in oral sex), and 55.3% (99/179) never used a condom for oral practices; the rest varied in the frequency of use. For receptive anal intercourse, lifetime consistent condom use was reported by 14.2% (25/176 participants engaging in anal sex); 48.3% (85/176) used a condom most of the time, 27.8% (49/176) rarely, and 9.6% (17/176) never.

The condom use data for various sexual practices over the previous 12 months according to sex of the participant are presented in Figure 1. As shown, consistent condom use over the previous 12 months was not frequent in women for all sexual practices, including anal sex; in contrast, one third of men reported consistent condom use during anal sex over the previous 12 months (Figure 1).

A history of at least one sexually transmitted infection (STI) was reported by 75.5% (151/200) of the participants, and 24.5% (49/200) reported never having had an STI. The most frequently reported STIs were syphilis 28% (55/200), perianal condyloma 25.5% (51/200), gonorrhea 16% (32/200), genital warts 16% (32/200), genital herpes 14.5% (29/200), pubic pediculosis 15% (29/200), urethritis 12% (24/200), hepatitis B 9.5% (19/200), and HPV 7.5% (15/200). Other reported STIs were hepatitis C and molluscum contagiosum (1.5% and 0.5%, respectively) (Figure 2). *Trichomonas* was not reported in the studied population.

Information concerning HIV viral load was obtained from 80.5% of the participants (161/200; data for 39 participants were not available). Of these, 61.5% (99/161) had an undetectable viral load or lower than 50 copies/mL; 26% (42/161) had a viral load ranging from 50 to 1000 copies/mL; and 12.4% (20/161) had viral loads ranging from 1000 to 20,000 copies/mL.

CD4+ counts were available for 161 participants (data for 39 participants were not available). Of these, 21.1% (34/161) had CD4+ counts lower than 200 cells/mm^3^; 41.6% (67/161) had >200 to 500 cells/mm^3^; and 37.26% (60/161) had CD4+ counts > 500 cells/mm^3^.

### 3.2. HPV Detection and Typing in Anal Canal and Cervix

All samples from men were obtained by self-sampling following the method established in the pilot, and all samples from women were obtained by staff. Anal samples were collected from all 200 participants (male and female) and cervix samples from all 31 female participants.

β-globin (quality control) was amplified in 194/200 of the anal samples, confirming that 97% of the samples were adequate for molecular testing. Those six samples (all from men) without β-globin amplification were considered inadequate and were not further considered for HPV detection or typing. The HPV L1 gene was detected in 116/194 of the adequate anal samples; therefore, 59.8% tested positive for HPV. According to the participants’ sex, 45.2% (14/31) of women and 62.6% (102/163) of men were HPV-positive.

The HPV typing method included HR-HPV types 16, 18, 31, 33, 45, 52, 58, 59, 35, 68, 39, and 66 and LR-HPV types 6/11, 42, 43, and 44. HPV genotypes were identified in 112 (96.6%) of the 116 positive samples. The remaining four HPV positives (3.4%) had non-identifiable genotypes (not included in the 18 genotype primers used). In general, the most frequent genotypes detected were LR-HPV 6/11 (36.3%) and HR-HPV33 (35.3%). HR-HPV 16, 18, 45, 51, and 66 were found at frequencies of 14.7, 13.7, 17.6, 13.7, and 18.6%, respectively. LR-HPV 43, 24, and 44 were also found (in 22.5, 16.7, 14.7% of the sample, respectively) (Figure 3).

Notably, multiple genotypes were detected in 80.2% of the anal samples, and 75% (85/112 identifiable genotypes) had at least one HR genotype.

HPV was detected in cervix samples from 60% of the women (18/30, with one inadequate sample after β-globin testing). All corresponded to at least one HR-HPV type, with HPV52 being the most frequent, found in 27.8% (5/18), followed by types 16, 18, 58, and 33, each found in 16.7% (3/18). Of the cervix samples from HPV-positive women, 89% (16/18) were infected with more than one genotype. In comparison, only 38.7% (12/31) were positive in the anal canal, and 10 had an infection in both anatomical regions, with a concordance of at least one genotype in most of these cases (9/10) (Table 1).

### 3.3. Anal Cytology

Available anal Pap smears and anal liquid-based cytology specimens were interpreted by an external cytopathology laboratory. In total, 197 anal cytopathology results were obtained for conventional anal Pap smears (three slides missing or broken) from 167 men (99%) and 30 women (97%), as follows: LSILs were found in 3 women and 33 men, and HSILs were found in 4 men.

For liquid-based cytology, a total of 108 results were available (92 missing) from 14 women and 94 men; among these, anal lesions were reported as follows: LSILs were found in 3 women and 33 men, and HSIL was found in 3 men.

To identify the agreement of the cytology methods, we included only a subset of the population. The group of 116 HPV PCR positives resulted in an overall good agreement (Table 2), though the inflammatory changes differed in a certain percentage. Fortunately, only four participants had HSILs as detected by either method, and in an additional patient HSILs were only detected by a conventional Pap smear; however, LSILs were slightly more frequently detected by liquid-based cytology (Table 3).

The genotypes present in HSIL cases were as follows: the first patient had a unique infection with HPV66; the second patient was infected with HPV52 and 33; the third patient had HPV 45, 33, 44, 51, and 66; and the fourth patient had HPV18, 6/11, 42, 68, 66, and 51.

### 3.4. Epidemiological Determinants

The possible association of HPV infection with epidemiological variables was analyzed, as shown in Table 3. The risk factors highly associated with HPV were having a new sexual partner for anal intercourse in the last 12 months and not using condom during anal intercourse in the last 12 months (*p* < 0.0001). Also significantly associated with HPV infection was having a detectable HIV viral load (>50 cc/mL) (*p* = 0.006). Smoking was marginally significant (*p* = 0.043). Age at first intercourse, CD4+ counts (<200 cells/mm^3^), and injectable drug use were not associated with HPV infection (Table 3).

Additional analyses showed that a detectable HIV viral load (>50 cc/mL) and low CD4+ counts (<200 cells/mm^3^) were not associated with anal intraepithelial lesions detected with either cytology technique (*p* > 0.05, data not shown).

## 4. Discussion

In this study, we analyzed the prevalence of HPV infection and cytological abnormalities in a group of men and women living with HIV in a region of Mexico with high HIV incidence, almost double the national level [8].

First, the importance of self-sampling for STI detection should be mentioned; it has been noted that the quality of such samples are comparable to samples obtained by trained personnel, and self-sampling has been proposed as an alternative approach to reach populations who do not perceive themselves to be at-risk [9]. In fact, self-sampling is comparable to medically collected samples for HPV detection [10]; in our study, the majority of samples were found to be adequate for PCR. It is worth mentioning that the self-sampling was performed in non-medical settings, which represents an advantage for performing research or even establishing screening strategies in contexts where medical offices or trained personnel are scarce. Self-sampling represents an advantage because of its high acceptability to participants; this approach was universally accepted by male participants and trans women (data not shown). In addition, due to the high detection rate of HPV in self-obtained anal samples, this has been proposed as an important tool for screening high-risk populations that would benefit those who refuse proctology examinations; additionally, it could represent the first step in algorithms for AIN screening [11].

However, in the case of cytology, the value of self-sampling has been discussed in relation to the lower quality of the specimens obtained. Therefore, it is important to explore and evaluate new strategies for patient instruction to determine if this could be a reliable sampling method, as reviewed by Yared et al. [10]. However, the studies included in this review were centered in developed countries only (United States, Canada, England, Ireland, and Australia). Our work demonstrated the viability of self-sampling in our setting and, although it could be optimized, established a basis for continued efforts in this area.

Studies analyzing the HPV infection of the anal canal have mainly focused on men who have sex with men (MSM), independently of their HIV status, as shown in a meta-analysis by Lin et al. [12], which identified 95 studies on the topic: 83% were performed with men and the rest in women, with only 11 studies identified in Latin America. This highlights the need to perform more studies in women, especially in Latin American regions. In our study, most participants (169/200) were men, and 94% of them reported sexual practices with men.

In Mexico, few studies on anal HPV are available. We reported close to 60% HPV+ positivity for anal canal samples in the total population, corresponding to 62.6% of men. In Mexico City (in the center of Mexico) a study in a group of 31 HIV+ MSM reported 74.19% (23/31) HPV anal infections [13], and a study in Chihuahua (in the north of Mexico) in a group of 69 HIV+ MSM reported 42% (29/69) [14].

The presence of HR-HPV types in the cited studies from Mexico in HIV+ MSM varied from 28 to 86% [13,14]. The variation between our study and previous reports from Mexico might be related to the method used for detection (hybrid capture, real-time PCR, and sequencing). In our study, we used end-point PCR, and HR infections represented 80% of all HPV-positive men.

A wide variety of genotypes were identified in the anal canal, but we had a particular interest in HR-HPV 16, 18, and 58, though their frequencies varied geographically (11–50%). A study performed in HIV+ MSM from Mexico reported 22.2%, 17.2%, and 13.9% prevalence for HPV 16, 58, and 18, respectively, and 48.8% for LR-HPV6/11 [15]. Another study reported a high prevalence (up to 93.1%) of HR-HPV infections, with HPV16 and 18 being the most frequent at 35.4% [16].

The results of this study are comparable to those of other international studies. In Hungary, HPV 16 was the most frequent at 42.5%, followed by types 18 (22.5%) and 58 (11.3%); the frequency of low-risk HPV6/11 was 49.1% in a population including HIV+ MSM [17]. In comparison, Rovelli et al. [18] analyzed 875 HIV+MSM in Italy, reporting a 27% frequency for HPV 16, 14% for HPV 18, and 11.3% for HPV 58, whilst genotypes 6/11 were less frequent (11%).

In fact, HR-HPV genotypes have been found in 85% of AIN cases, as shown in a meta-analysis by Machalek et al. [19], with HPV16 being the most frequent type at 75.9% in HIV+ MSM. In our study, less than one third of men had an infection with HPV16; however, we must consider that additional genotypes such as HPV18 and HPV58 have been associated with AIN development as in other cancers [5,20], and their risk should not be underestimated; the sum of those three genotypes represented c.a. 60% of men from our study group. Recently, Dr JM Palefsky, during his contribution to the STD Prevention Conference in Washington D.C. (September 2022), proposed that the screening of at-risk populations should include the genotype HPV16 only, based on experience obtained from large projects such as ANCHOR in the United States [21]. In our study, genotypes associated with HSIL cases were regarded as relevant, as they could be considered “less usual” than HPV16 (not found in these cases). HPV66 was found in three of these cases, HPV 33 in two, and HPV 18 in one. Longitudinal studies are needed to better describe the natural history of the lesions induced by these genotypes. In agreement with our observations, the 2018 meta-analysis by Lin et al. [12] and a 2020 report by Roberts et al. [22] deriving from the Australian SPANC project reported that HSILs in HIV+ men are more likely to be associated with genotypes other than HPV16 when compared to HIV- men.

Interestingly, in our study, the most frequent HR-HPV genotype in all men was HPV33 (33.3%), contrary to the abovementioned studies reporting low frequencies for this genotype (5–7%) [15,17,18]. In another study from Mexico using linear array, the most frequent genotypes were HPV62 and HPV81 [23]. In our study, these genotypes were not intentionally sought out, because they were not included in the employed method [7]. Despite the differences in the method used, this study agreed with our results in that multiple infections were very frequently found.

Regarding LR-HPV types, anal infections in HIV+ MSM are frequent and responsible for low-grade lesions, condyloma, and warts, which can be recurrent or also associated with other HR types [20,24]. In our study, a high percentage of participants reported a history of STIs: perianal condyloma was reported by 25%, and the most frequently found low-risk HPV genotypes were indeed HPV6 and 11 (61% altogether). Although these genotypes are not typically associated with malignancy, Siegenbeek van Heukelom et al. [24] reported the presence of HPV16 and 18 causing high-grade lesions in the area affected by perianal warts and condyloma; therefore, these genotypes should be observed carefully, as they can be indirectly indicative of AIN development.

Our study also included women; although this group was small, their participation was highly valuable as, to our knowledge, this is the first study to work with HIV+ women in Mexico. Previous studies have reported the prevalence of anal infections in HIV+ women to range from 31% in Zimbabwe to 90% in the United States [25,26]. In this study, we reported a positivity rate of 45.2%, comparable to other worldwide regions.

Women living with HIV and those with a history of cervical cancer are considered at a higher risk of developing anal pathologies; the anatomical proximity of the affected regions may play a role [27]. In a previous study from our research group involving 311 women with abnormal cervical cytology, HR-HPV (16, 18, 58, and 45) was found in 30.8% (96/311) of anal samples, and 11.25% had the same genotypes in both sites [28]. In the present study, 33.3% of our small group of HIV+ women had a HPV infection in both anatomical sites, with high agreement in terms of genotypes.

It is pertinent to mention that the genotypes included in the currently available HPV vaccine (Gardasil tetravalent version) are 6, 11, 16, and 18, and that this vaccine has been shown to be safe and immunogenic in up to 98% of HIV+ MSM [29]. In Mexican public health services, the vaccine is only offered to girls and is not administered to men. In the men from our study, the genotypes covered by this vaccine were detected with the following frequencies: HPV6/11 (44.4%), HPV16 (30.6%), and HPV18 (16%); therefore, it is important to implement vaccination as a primary prevention method in men, with regard to not only the genital/anal sites but also other cancer types such as head and neck [20].

The other genotypes identified in our population (31, 33, 45, 52, and 58) are covered by the Gardasil 9 nonavalent version, in addition to types 6, 11, 16, and 18. As of 2022, this vaccine is not available in Mexico; however, internationally, the vaccination of HIV+ men and women has shown benefits, and routine vaccination will hopefully become a reality in the near future for this vulnerable population.

Despite the possible cross-protection of the multivalent vaccines, it is important to note that infection with multiple genotypes in HIV+ cases is very frequent. In our study, the infection of the anal canal with multiple genotypes was frequent, accounting for 80% of all HPV-positive participants, similar to other studies in Latin America, where the rate of multiple infections ranged from 55 to 80.5% [15,30].

Risk factors for anal HPV infection include tobacco use, a higher number of sexual partners, sexual debut at an earlier age, and low condom use during anoreceptive intercourse [29]. In our study, tobacco use was frequent in men (32.3%) but not significantly associated; condoms were not used consistently by 86.7% of men during anoreceptive intercourse, representing the most important risk factor for exposure in this population. It is worth mentioning that condom use during oral sex was very low; the rates of HPV oral infection were not presented here and will be addressed in a separate study. In Mexico, data regarding oral cancer attributable to high-risk HPV are lacking.

Immunological and HIV therapy are related to HPV and lesion development, as evidenced in published meta-analysis [31]; however, in our study, CD4+ counts <200 cells/mm^3^ were not significantly associated with HPV infection or with cytological abnormalities (*p* > 0.05). On the contrary, detectable HIV RNA loads were indeed significantly associated with HPV infection (*p* = 0.006, OR = 3.05 (1.33–6.98), Table 3), but not with cytological abnormalities (*p* > 0.05, data not shown). This could have been an effect of the population size.

Anal cytology as a primary screening method for the detection of AIN is suggested as part of the triage of HIV+ MSM. A 2019 meta-analysis by Gonçalves et al. [32] including 4074 subjects, both HIV+ and HIV- women and men, evaluated its accuracy for the detection of AIN 2. The results for all subjects demonstrated values of 85.5 (83.3–97.6) and 51.5 (48.8–54.2) for sensitivity and specificity, respectively. The results were slightly different when only HIV+ MSM were included: sensitivity 85.5 (83.0–89.5), specificity 49.9 (45.2–53.6). In agreement with these results, the specificity could be improved with a HPV test [32].

In the development of our study, we faced the challenge that it is not common to find trained personnel for interpreting anal cytology in our region. Most HIV patients nationwide receive healthcare in public centers (CAPASITS—Centro Ambulatorio para la Atencion del VIH/SIDA e ITS), and the national guidelines for HIV care published by the Mexican health minister [33] include anal cytology upon the first visit by out-patients, but this is not performed in practice.

High-resolution anoscopy is considered the gold standard for the detection of anal cancer, and although official uniform international documents are lacking, several algorithms have been proposed [3]. Expert groups have established guidelines for defining standards for the detection of anal cancer precursor lesions, including the minimum requirements for anal cytology quality [34,35]. Moreover, most public hospitals lack specialists prepared to perform high-resolution anoscopy and related follow-ups, with such personnel only found in third-level hospitals. In our study, four cases of HSILs were identified among the participants that otherwise would not have been detected.

Finally, follow-ups for cases that are at risk of developing AIN is urged, including counseling and educational strategies to address risk factors.

## 5. Conclusions

In this study, we identified several risk factors associated with HPV infection in the anal canal and showed that HR-HPV types and co-infection with multiple types were frequent in the studied population; high-grade lesions, although infrequent, could be associated with genotypes considered uncommon in the general population and should not be underestimated. Women were of particularly high interest to us and should be prioritized in future research.

## Figures and Tables

**Figure 1 pathogens-12-00252-f001:**
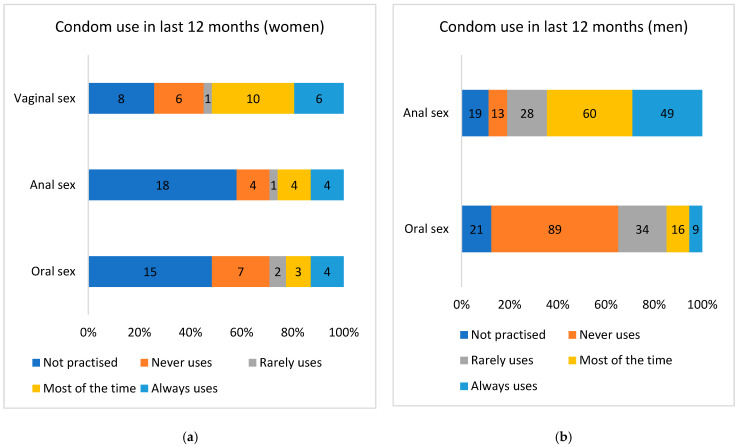
Condom use over the previous 12 months for different sexual practices (vaginal sex (receptive), oral sex (active), and anal sex (receptive)) in (**a**) women (n = 31 participants) and (**b**) men (n = 169 participants).

**Figure 2 pathogens-12-00252-f002:**
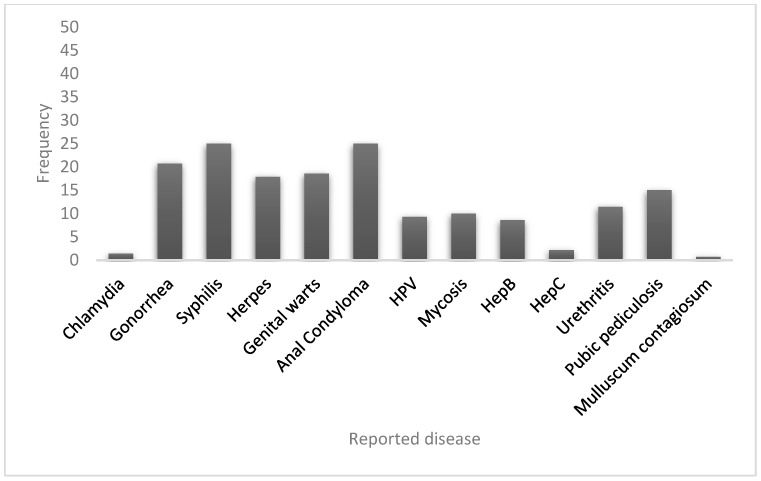
Frequency of reported history of sexually transmitted diseases, shown in percentages (n = 200 HIV+ participants).

**Figure 3 pathogens-12-00252-f003:**
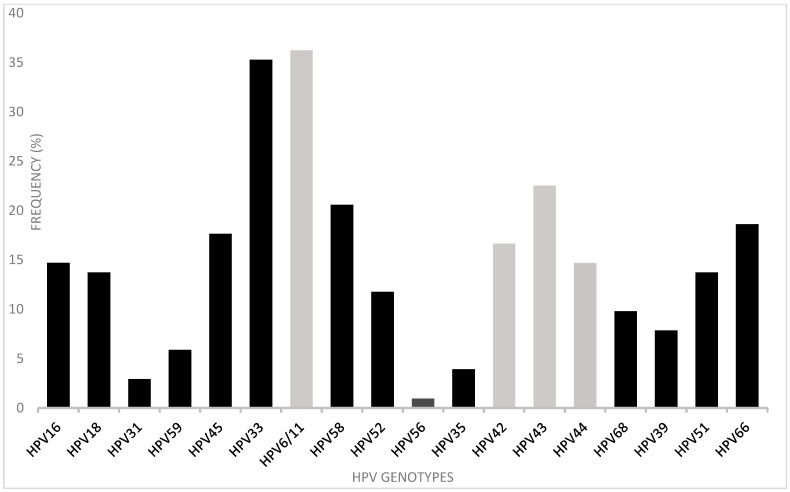
Frequency of genotypes present in anal samples from 112/116 HPV-positive participants, shown in percentages. HR types are shown by black bars, LR types by grey bars.

**Table 1 pathogens-12-00252-t001:** HPV genotypes in anal–cervix paired samples and agreement of identified genotypes. Samples at either anatomical site from only HPV+ women are shown.

Participant Code	Cervical HPV	Anal HPV	Cervical Genotypes	Anal Genotypes	Genotype Agreement *
VVH11	+	+	16, 59, 52, 33	16, 6/11, 66, 68	Yes
VVH22	+	+	18, 59, 6/11, 44	16, 6/11	Yes
VVH42	+	+	58, 44, 42	33, 42, 43, 44	Yes
VVH50	+	−	16	N/A	N/A
VVH51	+	+	16	16, 44, 68	Yes
VVH54	+	−	18, 45, 33, 58	N/A	N/A
VVH77	+	−	58, 44, 42	N/A	N/A
VVH114	+	+	33, 35, 39	33, 56, 66	Yes
VVH116	−	+	N/A	58	N/A
VVH117	+	+	43, 68	18, 43	Yes
VVH118	+	−	18, 56, 42, 39	N/A	N/A
VVH125	+	+	6/11, 42, 51	6/11, 51	Yes
VVH146	+	−	45, 44	N/A	N/A
VVH147	+	+	44, 66, 68	45	No
VVH150	+	+	45, 56, 51	56, 51	Yes
VVH156	+	−	45, 6/11	N/A	N/A
VVH160	+	−	6/11, 44	N/A	N/A
VVH183	inadequate	+	N/A	33, 42, 44	N/A
VVH187	+	+	52, 66	45, 52	Yes
VVH188	+	−	6/11	N/A	N/A

N/A, not applicable, + positive, − negative. * Agreement means at least one identical genotype present in both anatomical sites.

**Table 2 pathogens-12-00252-t002:** Conventional anal Pap smear and anal liquid-based cytology results from PCR HPV+ participants. LSIL (low-grade squamous intraepithelial lesion), HSIL (high-grade squamous intraepithelial lesion).

Anal Cytology Result	Pap Smear	Liquid-Based
LSIL, HPV infection	29 (25%)	33 (28.4%)
HSIL, cancer in situ	4 (3.4%)	3 (2.6%)
Inflammatory changes	40 (34.5%)	34 (29.3%)
Without abnormalities	41 (35.3%)	29 (25.0%)
Result not available	2 (1.7%)	17 (14.7%)
Total	116	116

**Table 3 pathogens-12-00252-t003:** Analysis of variables associated with HPV infection in the anal canal. OR, CI (odds ratio, confidence interval); X^2^ (chi-square test significance *p* < 0.05). Values shown in bold are significant.

Variables	HPV+	HPV−	OR (CI)	*p* Value (X^2^)
**Age at sexual debut**				
≤16 years	62/116 (53%)	43/78 (55%)	0.934 (0.525–1.663)	0.817
>16 years	54/116 (47%)	35/78 (45%)	1	
Total	116	78		
**Sexual partners** (anoreceptive intercourse, last 12 months)				
≥10	13/116 (10.3%)	42/78 (53.9%)	0.43 (0.19–0.99)	0.047
1–10	82/116 (70.7%)	5/78 (6.4%)	**23 (8.04–66.39)**	**<0.001**
None	22/116 (19%)	31/78 (39.7%)	1	
Total	116	78		
**Consistent condom use** (anoreceptive intercourse, last 12 months)				
No	58/94 (61.7%)	9/55 (16.4%)	**8.23 (3.60–18.82)**	**<0.001**
Yes	36/94 (38.3%)	46/55 (83.6%)	1	
Total (35 not applied)	94	55		
**HIV viral RNA load**				
≥50 cc/mL	31/93 (33.33%)	9/64 (14%)	**3.05 (1.33–6.98)**	**0.006**
<50 cc/mL	62/93 (66.66%)	55/64 (85.9%)	1	
Total (43 missing data)	93	64		
**CD4+ lymphocyte count**				
CD4 < 200 cells/mm^3^	23/93 (24.7%)	10/64 (15.6%)	1.77 (0.78–4.04)	0.168
CD4 > 200 cells/mm^3^	70/93 (75.3%)	54/64 (84.4%)	1	
Total (43 missing)	93	64		
**Active tobacco smoker**				
Yes	46/116 (39.65%)	20/78	**1.90 (1.015–3.57)**	**0.043**
No	70/116 (60.34%)	58/78	1	
Total	116	78		
**Injectable drug user**				
Yes	6	2	2.07 (0.40–10.54)	0.370
No	110	76	1	
Total	116	78		

## Data Availability

The data presented in this study are not public and are available from the corresponding author on reasonable request due to privacy.

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
