# Peer review of "Human Papillomavirus Genotypes Infecting the Anal Canal and Cervix in HIV+ Men and Women, Anal Cytology, and Risk Factors for Anal Infection"

_pathogens, 2023, doi:10.3390/pathogens12020252_

Round 1

Reviewer 1 Report

Dear authors 

Thank you for submitting your manuscript.

Please see comments of the paper on the attached document. 

Author Response

Please find attached a response to each of the reviewers’ comments. The revised manuscript shows the modified text with “Track changes” active mode.

The corresponding author.

Reviewer 2 Report

This manuscript explores the epidemiology of HPV in HIV+ individuals.

Certain genotypes of HPV are vaccine preventable. It would be worthwhile to comment on vaccine guidance in the study group for this area and which of the genotypes identified are included in the vaccine. It would also be good to add to the discussion whether vaccines would be helpful in the population studied here.

While it might be outside the scope of this research, considering the low rate of condom usage for oral sex, commentary on rates of oral cancer may be insightful in the introduction and/or discussion.

Figure 1 might be better presented as a table. It would also be interesting to see this data further striated by self-reported (monogamous -if this data was collected-) partnered relationships.

Self-sampling was used for swabbing in males but not females in this study. Presumably this is because clinicians collected anorectal and cervical swabs at the same time from female participants during pelvic exams. It would be better to state the reason for this difference in collection methods simply and clearly in the methods section and remove the references to differences in sampling methods elsewhere.

Minor comments

This manuscript would benefit for through English language grammar editing. There are many examples of awkwardly phrased statements and a few where it is difficult to discern the true meaning of the sentence.

Line 23: “titles” should be “titer”

Low-Grade Squamous Intraepithelial Lesion (LSIL) and High-Grade Squamous Intraepithelial Lesion (HSIL) are acronyms that should be identified when first used.

Line 60: “fly outs” should be “posters” or “fliers”

Please add the primer sequences for PGMY11/09 to the manuscript.

Line 92: please provide a citation for the “Bethesda nomenclature”

Line 110: “workers” should be “employed”

Line 111: Please remove the term “housewives”, “homemakers” is sufficient and avoids gender, orientation, and partner biases

Line 113: Did the survey differentiate between domestic partnerships and marriage? Partnered may be a better term here.

Line 115 & table 2: Please consider replacing “injectable drug users” with “persons who inject drugs”

Table 1: “Cervical HVP” should be “Cervical HPV”

Author Response

(The authors gave the same response as above.)
